# What Graph Neural Networks Cannot Learn: Depth vs Width

**Andreas Loukas**
École Polytechnique Fédérale Lausanne
`andreas.loukas@epfl.ch`

## Abstract

This paper studies the expressive power of graph neural networks falling within the message-passing framework ($\mathsf{GNN_{mp}}$). Two results are presented. First, $\mathsf{GNN_{mp}}$ are shown to be Turing universal under sufficient conditions on their depth, width, node attributes, and layer expressiveness. Second, it is discovered that $\mathsf{GNN_{mp}}$ can lose a significant portion of their power when their depth and width is restricted. The proposed impossibility statements stem from a new technique that enables the repurposing of seminal results from distributed computing and leads to lower bounds for an array of decision, optimization, and estimation problems involving graphs. Strikingly, several of these problems are deemed impossible unless the product of a $\mathsf{GNN_{mp}}$'s depth and width exceeds a polynomial of the graph size; this dependence remains significant even for tasks that appear simple or when considering approximation.

## 1 Introduction

A fundamental question in machine learning is to determine what a model *can* and *cannot* learn. In deep learning, there has been significant research effort in establishing expressivity results for feed-forward (Cybenko, 1989; Hornik et al., 1989; Lu et al., 2017) and recurrent neural networks (Neto et al., 1997), as well as more recently for Transformers and Neural GPUs (Pérez et al., 2019). We have also seen the first results studying the universality of graph neural networks, i.e., neural networks that take graphs as input. Maron et al. (2019b) derived a universal approximation theorem over invariant functions targeted towards deep networks whose layers are linear and equivariant to permutation of their input. Universality was also shown for equivariant functions and a particular shallow architecture (Keriven & Peyré, 2019).

Universality statements allow us to grasp the expressive power of models in the limit. In theory, given enough data and the right training procedure, a universal network will be able to solve any task that it is presented with. Nevertheless, the insight brought by such results can also be limited. Knowing that a sufficiently large network can be used to solve any problem does not reveal much about how neural networks should be designed in practice. It also cannot guarantee that said network will be able to solve a given task given a particular training procedure, such as stochastic gradient descent.

On the other hand, it might be easier to obtain insights about models by studying their limitations. After all, the knowledge of what *cannot* be computed (and thus learned) by a network of specific characteristics applies independently of the training procedure. Further, by helping us comprehend the difficulty of a task in relation to a model, impossibility results can yield practical advice on how to select model hyperparameters. Take, for instance, the problem of graph classification. Training a graph classifier entails identifying what constitutes a class, i.e., finding properties shared by graphs in one class but not the other, and then deciding whether new graphs abide to said learned properties. However, if the aforementioned decision problem is shown to be impossible by a graph neural network of certain depth then we can be certain that the same network will not learn how to classify a sufficiently diverse test set correctly, independently of which learning algorithm is employed. We should, therefore, focus on networks deeper that the lower bound when performing experiments.

| problem | bound | problem | bound |
|---------|-------|---------|-------|
| cycle detection (odd) | $dw = \Omega(n/\log n)$ | shortest path | $d\sqrt{w} = \Omega(\sqrt{n}/\log n)$ |
| cycle detection (even) | $dw = \Omega(\sqrt{n}/\log n)$ | max. indep. set | $dw = \Omega(n^2/\log^2 n)$ for $w = O(1)$ |
| subgraph verification* | $d\sqrt{w} = \Omega(\sqrt{n}/\log n)$ | min. vertex cover | $dw = \Omega(n^2/\log^2 n)$ for $w = O(1)$ |
| min. spanning tree | $d\sqrt{w} = \Omega(\sqrt{n}/\log n)$ | perfect coloring | $dw = \Omega(n^2/\log^2 n)$ for $w = O(1)$ |
| min. cut | $d\sqrt{w} = \Omega(\sqrt{n}/\log n)$ | girth 2-approx. | $dw = \Omega(\sqrt{n}/\log n)$ |
| diam. computation | $dw = \Omega(n/\log n)$ | diam. $^3/_2$-approx. | $dw = \Omega(\sqrt{n}/\log n)$ |

Table 1: Summary of main results. Subgraph verification* entails verifying one of the following predicates for a given subgraph: is connected, contains a cycle, forms a spanning tree, is bipartite, is a cut, is an $s$-$t$ cut. All problems are defined in Appendix A.

## 1.1 MAIN RESULTS

This paper studies the expressive power of message-passing graph neural networks ($\mathsf{GNN_{mp}}$) (Gilmer et al., 2017). This model encompasses several state-of-the-art networks, including GCN (Kipf & Welling, 2016), gated graph neural networks (Li et al., 2015), molecular fingerprints (Duvenaud et al., 2015), interaction networks (Battaglia et al., 2016), molecular convolutions (Kearnes et al., 2016), among many others. Networks using a global state (Battaglia et al., 2018) or looking at multiple hops per layer (Morris et al., 2019; Liao et al., 2019; Isufi et al., 2020) are not directly $\mathsf{GNN_{mp}}$, but they can often be re-expressed as such. The provided contributions are two-fold:

**I. What $\mathsf{GNN_{mp}}$ can compute.** Section 3 derives sufficient conditions such that a $\mathsf{GNN_{mp}}$ can compute any function on its input that is computable by a Turing machine. This result compliments recent universality results (Maron et al., 2019b; Keriven & Peyré, 2019) that considered approximation (rather than computability) over specific classes of functions (permutation invariant and equivariant) and particular architectures. The claim follows in a straightforward manner by establishing the equivalence of $\mathsf{GNN_{mp}}$ with $\mathsf{LOCAL}$ (Angluin, 1980; Linial, 1992; Naor & Stockmeyer, 1993), a classical model in distributed computing that is itself Turing universal. In a nutshell, $\mathsf{GNN_{mp}}$ are shown to be universal if four strong conditions are met: there are enough layers of sufficient expressiveness and width, and nodes can uniquely distinguish each other. Since Turing universality is a strictly stronger property than universal approximation, Chen et al. (2019)'s argument further implies that a Turing universal $\mathsf{GNN_{mp}}$ can solve the graph isomorphism problem (a sufficiently deep and wide network can compute the isomorphism class of its input).

**II. What $\mathsf{GNN_{mp}}$ cannot compute (and thus learn).** Section 4 analyses the implications of restricting the depth $d$ and width $w$ of $\mathsf{GNN_{mp}}$ that do not use a readout function. Specifically, it is proven that $\mathsf{GNN_{mp}}$ lose a significant portion of their power when the product $dw$, which I call *capacity*, is restricted. The analysis relies on a new technique that enables repurposing impossibility results from the context of distributed computing to the graph neural network setting. Specifically, lower bounds for the following problems are presented: (i) detecting whether a graph contains a cycle of specific length; (ii) verifying whether a given subgraph is connected, contains a cycle, is a spanning tree, is bipartite, is a simple path, corresponds to a cut or Hamiltonial cycle; (iii) approximating the shortest path between two nodes, the minimum cut, and the minimum spanning tree; (iv) finding a maximum independent set, a minimum vertex cover, or a perfect coloring; (v) computing or approximating the diameter and girth. The bounds are summarized in Table 1 and the problem definitions can be found in Appendix A. Section 5 presents some empirical evidence of the theory.

Though formulated in a graph-theoretic sense, the above problems are intimately linked to machine learning on graphs. Detection, verification, and computation problems are relevant to classification: knowing what properties of a graph a $\mathsf{GNN_{mp}}$ cannot see informs us also about which features of a graph can it extract. Further, there have been attempts to use $\mathsf{GNN_{mp}}$ to devise heuristics for graph-based optimization problems (Khalil et al., 2017; Battaglia et al., 2018; Li et al., 2018; Joshi et al., 2019; Bianchi et al., 2019), such as the ones discussed above. The presented results can then be taken as a worst-case analysis for the efficiency of $\mathsf{GNN_{mp}}$ in such endeavors.

## 1.2 DISCUSSION

The results of this paper carry several intriguing implications. *To start with, it is shown that the capacity $dw$ of a $\mathsf{GNN_{mp}}$ plays a significant role in determining its power.* Solving many problems

is shown to be impossible unless $dw = \tilde{\Omega}(n^\delta)$, where $\delta \in [1/2, 2]$, $n$ is the number of nodes of the graph, and $f(n) = \tilde{\Omega}(g(n))$ is interpreted as $f(n)$ being, up to logarithmic factors, larger than $g(n)$ as $n$ grows. This reveals a direct trade-off between the depth and width of a graph neural network. *Counter-intuitively, the dependence on $n$ can be significant even if the problem appears local in nature or one only looks for approximate solutions.* For example, detecting whether $G$ contains a short cycle of odd length cannot be done unless $dw = \tilde{\Omega}(n)$. Approximation helps, but only to a limited extent; computing the graph diameter requires $dw = \tilde{\Omega}(n)$ and this reduces to $dw = \tilde{\Omega}(\sqrt{n})$ for any $3/2$-factor approximation. Further, it is impossible to approximate within *any* constant factor the shortest path, the minimum cut, and the minimum spanning tree, all three of which have polynomial-time solutions, unless $d\sqrt{w} = \tilde{\Omega}(\sqrt{n})$. *Finally, for truly hard problems, the capacity may even need to be super-linear on $n$.* Specifically, it is shown that, even if the layers of the $\mathsf{GNN_{mp}}$ are allowed to take exponential time, solving certain NP-hard problems necessitates $d = \tilde{\Omega}(n^2)$ depth for any constant-width network.

**Relation to previous impossibility results.** In contrast to universality (Maron et al., 2019b; Keriven & Peyré, 2019), the limitations of $\mathsf{GNN_{mp}}$ have been much less studied. In particular, the bounds presented here are the first impossibility results that (i) explicitly connect $\mathsf{GNN_{mp}}$ properties (depth and width) with graph properties and that (ii) go beyond isomorphism by addressing decision, optimization, and estimation graph problems. Three main directions of related work can be distinguished. First, Dehmamy et al. (2019) bounded the ability of graph convolutional networks (i.e., $\mathsf{GNN_{mp}}$ w/o messaging functions) to compute specific polynomial functions of the adjacency matrix, referred to as graph moments by the authors. Second, Xu et al. (2018) and Morris et al. (2019) established the equivalence of *anonymous* $\mathsf{GNN_{mp}}$ (those that do not rely on node identification) to the Weisfeiler-Lehman (WL) graph isomorphism test. The equivalence implies that anonymous networks are blind to the many graph properties that WL cannot see: e.g., any two regular graphs with the same number of nodes are identical from the perspective of the WL test (Arvind et al., 2015; Kiefer et al., 2015). Third, in parallel to this work, Sato et al. (2019) utilized a connection to $\mathsf{LOCAL}$ to derive impossibility results for the ability of a class of novel *partially-labeled* $\mathsf{GNN_{mp}}$ to find good approximations for three NP-hard optimization problems. Almost all of the above negative results occur due to nodes being unable to distinguish between neighbors at multiple hops (see Appendix D). With discriminative attributes $\mathsf{GNN_{mp}}$ become significantly more powerful (without necessarily sacrificing permutation in/equivariance). Still, as this work shows, even in this setting certain problems remain impossible when the depth and width of the $\mathsf{GNN_{mp}}$ is restricted. For instance, though cycles can be detected (something impossible in anonymous networks (Xu et al., 2018; Morris et al., 2019)), even for short cycles one now needs $dw = \tilde{\Omega}(n)$. Further, in contrast to Sato et al. (2019), an approximation ratio below 2 for the minimum vertex cover is not impossible, but necessitates $dw = \tilde{\Omega}(n^2)$.

**Limitations.** First, all lower bounds are of a worst-case nature: a problem is deemed impossible if there exists a graph for which it cannot be solved. The discovery of non worst-case capacity bounds remains an open problem. Second, rather than taking into account specific parametric functions, each layer is assumed to be sufficiently powerful to compute any function of its input. This strong assumption does not significantly limit the applicability of the results, simply because all lower bounds that hold with universal layers also apply to those that are limited computationally. Lastly, it will be assumed that nodes can uniquely identify each other. Node identification is compatible with permutation invariance/equivariance as long as the network output is asked to be invariant to the particular way the ids have been assigned. In the literature, one-hot encoded node ids are occasionally useful (Kipf & Welling, 2016; Berg et al., 2017). When attempting to learn functions across multiple graphs, ids should be ideally substituted by sufficiently discriminative node attributes (attributes that uniquely identify each node within each receptive field it belongs to can serve as ids). Nevertheless, similar to the unbounded computation assumption, if a problem cannot be solved by a graph neural network in the studied setting, it also cannot be solved without identifiers and discriminative attributes. Thus, the presented lower bounds also apply to partially and fully anonymous networks.

**Notation.** I consider connected graphs $G = (\mathcal{V}, \mathcal{E})$ consisting of $n = |\mathcal{V}|$ nodes. The edge going from $v_j$ to $v_i$ is written as $e_{i \leftarrow j}$ and it is asserted that if $e_{i \leftarrow j} \in \mathcal{E}$ then also $e_{j \leftarrow i} \in \mathcal{E}$. The neighborhood $\mathcal{N}_i$ of a node $v_i \in \mathcal{V}$ consists of all nodes $v_j$ for which $e_{i \leftarrow j} \in \mathcal{E}$. The degree of $v_i$ is denoted by $\deg_i$, $\Delta$ is the maximum degree of all nodes and the graph diameter $\delta_G$ is the length of the longest

shortest path between any two nodes. In the self-loop graph $G^* = (\mathcal{V}, \mathcal{E}^*)$, the neighborhood set of $v_i$ is given by $\mathcal{N}_i^* = \mathcal{N}_i \cup v_i$.

## 2 THE GRAPH NEURAL NETWORK COMPUTATIONAL MODEL

Graph neural networks are parametric and differentiable learning machines. Their input is usually an attributed graph $G_a = (G, (a_i : v_i \in \mathcal{V}), (a_{i \leftarrow j} : e_{i \leftarrow j} \in \mathcal{E}))$, where vectors $a_1, \ldots, a_n$ encode relevant node attributes and $a_{i \leftarrow j}$ are edge attributes, e.g., encoding edge direction.

Model 1 formalizes the graph neural network operation by placing it in the message passing model (Gilmer et al., 2017). The computation proceeds in layers, within which a message $m_{i \leftarrow j}$ is passed along each directed edge $e_{i \leftarrow j} \in \mathcal{E}$ going from $v_j$ to $v_i$ and each node updates its internal representation by aggregating its state with the messages sent by its incoming neighbors $v_j \in \mathcal{N}_i$. The network output can be either of two things: a vector $x_i$ for each node $v_i$ or a single vector $x_G$ obtained by combining the representations of all nodes using a readout function. Vectors $x_i/x_G$ could be scalars (node/graph regression), binary variables (node/graph classification) or multi-dimensional (node/graph embedding). I use the symbols $\mathsf{GNN}_{mp}^n$ and $\mathsf{GNN}_{mp}^g$ to distinguish between models that return a vector per node and one per graph, respectively.

---

**Computational model 1** Message passing graph neural network ($\mathsf{GNN}_{mp}$)

---

**Initialization:** Set $x_i^{(0)} = a_i$ for all $v_i \in \mathcal{V}$.
**for** layer $\ell = 1, \ldots, d$ **do**
    **for** every edge $e_{i \leftarrow j} \in \mathcal{E}^*$ (in parallel) **do**

$$m_{i \leftarrow j}^{(\ell)} = \text{MSG}_\ell \left( x_i^{(\ell-1)}, x_j^{(\ell-1)}, v_i, v_j, a_{i \leftarrow j} \right)$$

    **for** every node $v_i \in \mathcal{V}$ (in parallel) **do**

$$x_i^{(\ell)} = \text{UP}_\ell \Big( \sum_{v_j \in \mathcal{N}_i^*} m_{i \leftarrow j}^{(\ell)} \Big)$$

Set $x_i = x_i^{(d)}$.
**return** Either $x_i$ for every $v_i \in \mathcal{V}$ ($\mathsf{GNN}_{mp}^n$) or $x_G = \text{READ} \left( \{ x_i : v_i \in \mathcal{V} \} \right)$ ($\mathsf{GNN}_{mp}^g$).

---

The operation of a $\mathsf{GNN}_{mp}$ is primarily determined by the *messaging*, *update*, and *readout* functions. I assume that $\text{MSG}_\ell$ and $\text{UP}_\ell$ are general functions that act on intermediate node representations and node ids (the notation is overloaded such that $v_i$ refers to both the $i$-th node as well as its unique id). As is common in the literature (Lu et al., 2017; Battaglia et al., 2018), these functions are instantiated by feed-forward neural networks. Thus, by the universal approximation theorem and its variants (Cybenko, 1989; Hornik et al., 1989), they can approximate any general function that maps vectors onto vectors, given sufficient depth and/or width. Function READ is useful when one needs to retrieve a representation that is invariant of the number of nodes. The function takes as an input a *multiset*, i.e., a set with possibly repeating elements, and returns a vector. Commonly, READ is chosen to be a dimension squashing operator, such as a sum or a histogram, followed by a feed-forward neural network (Xu et al., 2018; Seo et al., 2019).

**Depth and width.** The depth $d$ is equal to the number of layers of the network. Larger depth means that each node has the opportunity to learn more about the rest of the graph (i.e., it has a larger receptive field). The width $w$ of a $\mathsf{GNN}_{mp}$ is equal to the largest dimension of state $x_i^{(l)}$ over all layers $l$ and nodes $v_i \in \mathcal{V}$. Since nodes need to be able to store their own unique ids, in the following it is assumed that each variable manipulated by the network is represented in finite-precision using $p = \Theta(\log n)$ bits (though this is not strictly necessary for the analysis).

## 3 SUFFICIENT CONDITIONS FOR TURING UNIVERSALITY

This section studies what graph neural networks can compute. It is demonstrated that, even without readout function, a network is computationally universal[1] if it has enough layers of sufficient

---

[1]It can compute anything that a Turing machine can compute when given an attributed graph as input.

width, nodes can uniquely distinguish each other, and the functions computed within each layer are sufficiently expressive. The derivation entails establishing that $\mathsf{GNN}_{mp}^n$ is equivalent to $\mathsf{LOCAL}$, a classical model used in the study the distributed algorithms that is itself Turing universal.

## 3.1 THE LOCAL COMPUTATIONAL MODEL

A fundamental question in theoretical computer science is determining what can and cannot be computed efficiently by a distributed algorithm. The $\mathsf{LOCAL}$ model, initially studied by Angluin (1980), Linial (1992), and Naor & Stockmeyer (1993), provides a common framework for analyzing the effect of local decision. Akin to $\mathsf{GNN}_{mp}$, in $\mathsf{LOCAL}$ a graph plays a double role: it is both the input of the system and captures the network topology of the distributed system that solves the problem. In this spirit, the nodes of the graph are here both the machines where computation takes place as well as the variables of the graph-theoretic problem we wish to solve—similarly, edges model communication links between machines as well as relations between nodes. Each node $v_i \in \mathcal{V}$ is given a problem-specific local input and has to produce a local output. The input contains necessary the information that specifies the problem instance. All nodes execute the same algorithm, they are fault-free, and they are provided with unique identifiers.

A pseudo-code description is given in Model 2. Variables $s_i^{(l)}$ and $s_{i\leftarrow j}^{(l)}$ refer respectively to the state of $v_i$ in round $l$ and to the message sent by $v_j$ to $v_i$ in the same round. Both are represented as strings. The computation starts simultaneously and unfolds in synchronous rounds $l = 1, \ldots, d$. Three things can occur within each round: each node receives a string of *unbounded* size from its incoming neighbors; each node updates its internal state by performing some local computation; and each node sends a string to every one of its outgoing neighbors. Functions $\mathrm{ALG}_l^1$ and $\mathrm{ALG}_l^2$ are algorithms computed locally by a Turing machine running on node $v_i$. Before any computation is done, each node $v_i$ is aware of its own attribute $a_i$ as well as of all edge attributes $\{a_{i\leftarrow j} : v_j \in \mathcal{N}_i^*\}$.

---

**Computational model 2** LOCAL (computed distributedly by each node $v_i \in \mathcal{V}$).

**Initialization:** Set $s_{i\leftarrow i}^{(0)} = (a_i, v_i)$ and $s_{i\leftarrow j}^{(0)} = (a_j, v_j)$ for all $e_{i\leftarrow j} \in \mathcal{E}$.
**for** round $\ell = 1, \ldots, d$ **do**
    Receive $s_{i\leftarrow j}^{(\ell-1)}$ from $v_j \in \mathcal{N}_i^*$, compute
$$s_i^{(\ell)} = \mathrm{ALG}_\ell^1 \left( \left\{ \left( s_{i\leftarrow j}^{(\ell-1)}, a_{i\leftarrow j} \right) : v_j \in \mathcal{N}_i^* \right\}, v_i \right),$$
    and send $s_{j\leftarrow i}^{(\ell)} = \mathrm{ALG}_\ell^2 \left( s_i^{(\ell)}, v_i \right)$ to $v_j \in \mathcal{N}_i^*$.
**return** $s_i^{(d)}$

---

In $\mathsf{LOCAL}$, there are no restrictions on how much information a node can send at every round. Asserting that each message $s_{i\leftarrow j}^{(\ell)}$ is at most $b$ bits yields the $\mathsf{CONGEST}$ model (Peleg, 2000).

## 3.2 TURING UNIVERSALITY

The reader might have observed that $\mathsf{LOCAL}$ resembles closely $\mathsf{GNN}_{mp}^n$ in its structure, with only a few minor differences: firstly, whereas a $\mathsf{LOCAL}$ algorithm $\mathsf{A}$ may utilize messages in any way it chooses, a $\mathsf{GNN}_{mp}^n$ network $\mathsf{N}$ always sums received messages before any local computation. The two models also differ in the arguments of the messaging function and the choice of information representation (string versus vector). Yet, as the following theorem shows, the differences between $\mathsf{GNN}_{mp}^n$ and $\mathsf{LOCAL}$ are inconsequential when seen from the perspective of their expressive power:

**Theorem 3.1** (Equivalence). *Let $\mathsf{N}_\ell(G_a)$ be the binary representation of the state $(x_1^{(\ell)}, \ldots, x_n^{(\ell)})$ of a $\mathsf{GNN}_{mp}^n$ network $\mathsf{N}$ and $\mathsf{A}_\ell(G_a) = (s_1^{(\ell)}, \ldots, s_n^{(\ell)})$ that of a $\mathsf{LOCAL}$ algorithm $\mathsf{A}$. If $\mathrm{MSG}_\ell$ and $\mathrm{UP}_\ell$ are Turing complete functions, then, for any algorithm $\mathsf{A}$ there exists $\mathsf{N}$ (resp. for any $\mathsf{N}$ there exists $\mathsf{A}$) such that*

$$\mathsf{A}_\ell(G_a) = \mathsf{N}_\ell(G_a) \quad \text{for every layer } \ell \text{ and } G_a \in \mathcal{G}_a,$$

*where $\mathcal{G}_a$ is the set of all attributed graphs.*

This equivalence enables us to reason about the power of $GNN_{mp}^n$ by building on the well-studied properties of LOCAL. In particular, it is well known in distributed computing that, as long as the number of rounds $d$ of a distributed algorithm is larger than the graph diameter $\delta_G$, every node in a LOCAL can effectively make decisions based on the entire graph (Linial, 1992). Together with Theorem 3.1, the above imply that, if computation and memory are not an issue, one may construct a $GNN_{mp}^n$ that effectively computes *any* computable function w.r.t. its input.

**Corollary 3.1.** *$GNN_{mp}^n$ can compute any Turing computable function over connected attributed graphs if the following conditions are jointly met: each node is uniquely identified; $\text{MSG}_l$ and $\text{UP}_l$ are Turing-complete for every layer $\ell$; the depth is at least $d \geq \delta_G$ layers; and the width is unbounded.*

Why is this result relevant? From a cursory review, it might seem that universality is an abstract result with little implication to machine learning architects. After all, the utility of a learning machine is usually determined not with regards to its expressive power but with its ability to generalize to unseen examples. Nevertheless, it can be argued that universality is an essential property of a good learning model. This is for two main reasons: *First, universality guarantees that the learner does not have blind-spots in its hypothesis space.* No matter how good the optimization algorithm is, how rich the dataset, and how overparameterized the network is, there will always be functions which a non universal learner cannot learn. *Second, a universality result provides a glimpse on how the size of the learner's hypothesis space is affected by different design choices.* For instance, Corollary 3.1 puts forth four necessary conditions for universality: the $GNN_{mp}^n$ should be sufficiently deep and wide, nodes should be able to uniquely and consistently identify each other, and finally, the functions utilized in each layer should be sufficiently complex. The following section delves further into the importance of two of these universality conditions. It will be shown that $GNN_{mp}^n$ lose a significant portion of their power when the depth and width conditions are relaxed.

**The universality of $GNN_{mp}^g$.** Though a universality result could also be easily derived for networks with a readout function, the latter is not included as it deviates from how graph neural networks are meant to function: given a sufficiently powerful readout function, a $GNN_{mp}^g$ of $d = 1$ depth and $O(\Delta)$ width can be used to compute any Turing computable function. The nodes should simply gather one hop information about their neighbors; the readout function can then reconstruct the problem input based on the collective knowledge and apply any computation needed.

## 4 IMPOSSIBILITY RESULTS AS A FUNCTION OF DEPTH AND WIDTH

This section analyzes the effect of depth and width in the expressive power of $GNN_{mp}^n$. Specifically, I will consider problems that cannot be solved by a network of a given depth and width.

To be able to reason in terms of width, it will be useful to also enforce that the message size in LOCAL at each round is at most $b$ bits. This model goes by the name CONGEST in the distributed computing literature (Peleg, 2000). In addition, it will be assumed that nodes do not have access to a random generator. With this in place, the following theorem shows us how to translate impossibility results from CONGEST to $GNN_{mp}^n$:

**Theorem 4.1.** *If a problem $P$ cannot be solved in less than $d$ rounds in CONGEST using messages of at most $b$ bits, then $P$ cannot be solved by a $GNN_{mp}^n$ of width $w \leq (b - \log_2 n)/p = O(b/\log n)$ and depth $d$.*

The $p = \Theta(\log n)$ factor corresponds to the length of the binary representation of every variable—the precision needs to depend logarithmically on $n$ for the node ids to be unique. With this result in place, the following sections re-state several known lower bounds in terms of a $GNN_{mp}^n$'s depth and width.

### 4.1 IMPOSSIBILITY RESULTS FOR DECISION PROBLEMS

I first consider problems where one needs to decide whether a given graph satisfies a certain property (Feuilloley & Fraigniaud, 2016). Concretely, given a decision problem $P$ and a graph $G$, the $GNN_{mp}^n$ should output $x_i \in \{true, false\}$ for all $v_i \in \mathcal{V}$. The network then accepts the premise if the logical conjunction of $\{x_1, \ldots, x_n\}$ is *true* and rejects it otherwise. Such problems are intimately connected to graph classification: classifying a graph entails identifying what constitutes a class from some training set *and* using said learned definition to decide the label of graphs sampled from

the test set. Instead, I will suppose that the class definition is available to the classifier and I will focus on the corresponding decision problem. As a consequence, every lower bound presented below for a decision problem must also be respected by a $\mathsf{GNN}^n_{mp}$ classifier that attains zero error on the corresponding graph classification problem.

**Subgraph detection.** In this type of problems, the objective is to decide whether $G$ contains a subgraph belonging to a given family. I focus specifically on detecting whether $G$ contains a cycle $C_k$, i.e., a simple undirected graph of $k$ nodes each having exactly two neighbors. As the following result shows, even with ids, cycle detection remains relatively hard:

**Corollary 4.1** (Repurposed from (Drucker et al., 2014; Korhonen & Rybicki, 2018)). *There exists graph $G$ on which every $\mathsf{GNN}^n_{mp}$ of width $w$ requires depth at least $d = \Omega(\sqrt{n}/(w \log n))$ and $d = \Omega(n/(w \log n))$ to detect if $G$ contains a cycle $C_k$ for even $k \geq 4$ and odd $k \geq 5$, respectively.*

Whereas an anonymous $\mathsf{GNN}_{mp}$ cannot detect cycles (e.g., distinguish between two $C_3$ vs one $C_6$ (Maron et al., 2019a)), it seems that with ids the product of depth and width should exhibit an (at least) linear dependence on $n$. The intuition behind this bound can be found in Appendix C and empirical evidence in support of the theory are presented in Section 5.

**Subgraph verification.** Suppose that the network is given a subgraph $H = (\mathcal{V}_H, \mathcal{E}_H)$ of $G$ in its input. This could, for instance, be achieved by selecting the attributes of each node and edge to be a one-hot encoding of their membership on $\mathcal{V}_H$ and $\mathcal{E}_H$, respectively. The question considered is whether the neural network can verify a certain property of $H$. More concretely, does a graph neural network exist that can successfully verify $H$ as belonging to a specific family of graphs w.r.t. $G$? In contrast to the standard decision paradigm, here every node should reach the same decision—either accepting or rejecting the hypothesis. The following result is a direct consequence of the seminal work by Sarma et al. (2012):

**Corollary 4.2** (Repurposed from (Sarma et al., 2012)). *There exists a graph $G$ on which every $\mathsf{GNN}^n_{mp}$ of width $w$ requires depth at least $d = \Omega(\sqrt{\frac{n}{w \log^2 n}} + \delta_G)$ to verify if some subgraph $H$ of $G$ is connected, contains a cycle, forms a spanning tree of $G$, is bipartite, is a cut of $G$, or is an s-t cut of $G$. Furthermore, the depth should be at least $d = \Omega\left(\left(\frac{n}{w \log n}\right)^{\gamma} + \delta_G\right)$ with $\gamma = \frac{1}{2} - \frac{1}{2(\delta_{G'} - 1)}$ to verify if $H$ is a Hamiltonian cycle or a simple path.*

Therefore, even if one knows where to look in $G$, verifying whether a given subgraph meets a given property can be non-trivial, and this holds for several standard graph-theoretic properties. For instance, if we constrain ourselves to networks of constant width, detecting whether a subgraph is connected can, up to logarithmic factors, require $\Omega(\sqrt{n})$ depth in the worst case.

## 4.2 Impossibility results for optimization problems

I turn my attention to the problems involving the exact or approximate optimization of some graph-theoretic objective function. From a machine learning perspective, the considered problems can be interpreted as node/edge classification problems: each node/edge is tasked with deciding whether it belongs to the optimal set or not. Take, for instance, the maximum independent set, where one needs to find the largest cardinality node set, such that no two of them are adjacent. Given only information identifying nodes, $\mathsf{GNN}^n_{mp}$ will be asked to classify each node as being part of the maximum independent set or not.

**Polynomial-time problems.** Let me first consider three problems that possess known polynomial-time solutions. To make things easier for the $\mathsf{GNN}^n_{mp}$, I relax the objective and ask for an approximate solution rather than optimal. An algorithm (or neural network) is said to attain an $\alpha$-approximation if it produces a feasible output whose utility is within a factor $\alpha$ of the optimal. Let OPT be the utility of the optimal solution and ALG that of the $\alpha$-approximation algorithm. Depending on whether the problem entails minimization or maximization, the ratio ALG/OPT is at most $\alpha$ and at least $1/\alpha$, respectively.

According to the following corollary, it is non-trivial to find good approximate solutions:

**Corollary 4.3** (Repurposed from (Sarma et al., 2012; Ghaffari & Kuhn, 2013)). *There exists graphs $G$ and $G'$ of diameter $\delta_G = \Theta(\log n)$ and $\delta_{G'} = O(1)$ on which every $\mathsf{GNN}^n_{mp}$ of width $w$ requires*

*depth at least $d = \Omega(\sqrt{\frac{n}{w \log^2 n}})$ and $d' = \Omega((\frac{n}{w \log n})^\gamma)$ with $\gamma = \frac{1}{2} - \frac{1}{2(\delta_{G'}-1)}$, respectively, to approximate within any constant factor: the minimum cut problem, the shortest s-t path problem, or the minimum spanning tree problem.*

Thus, even for simple problems (complexity-wise), in the worst case a constant width $\mathsf{GNN}_{mp}^n$ should be almost $\Omega(\sqrt{n})$ deep even if the graph diameter is exponentially smaller than $n$.

**NP-hard problems.** So what about truly hard problems? Clearly, one cannot expect a $\mathsf{GNN}_{mp}$ to solve an NP-hard time in polynomial time[2]. However, it might be interesting as a thought experiment to consider a network whose layers take exponential time on the input size—e.g., by selecting the $\textsc{Msg}_l$ and $\textsc{Up}_l$ functions to be feed-forward networks of exponential depth and width. Could one ever expect such a $\mathsf{GNN}_{mp}^n$ to arrive at the optimal solution?

The following corollary provides necessary conditions for three well-known NP-hard problems:

**Corollary 4.4** (Repurposed from (Censor-Hillel et al., 2017)). *There exists a graph $G$ on which every $\mathsf{GNN}_{mp}^n$ of width $w = O(1)$ requires depth at least $d = \Omega(n^2/\log^2 n)$ to solve: the minimum vertex cover problem; the maximum independent set problem; the perfect coloring problem.*

Thus, even if each layer is allowed to take exponential time, the depth should be quadratically larger than the graph diameter $\delta_G = O(n)$ to have a chance of finding the optimal solution. Perhaps disappointingly, the above result suggests that it may not be always possible to exploit the distributed decision making performed by $\mathsf{GNN}_{mp}$ architectures to find solutions faster than classical (centralized) computational paradigms.

### 4.3 Impossibility results for estimation problems

Finally, I will consider problems that involve the computation or estimation of some real function that takes as an input the graph and attributes. The following corollary concerns the computation of two well-known graph invariants: the *diameter* $\delta_G$ and the *girth*. The latter is defined as the length of the shortest cycle and is infinity if the graph has no cycles.

**Corollary 4.5** (Repurposed from (Frischknecht et al., 2012)). *There exists a graph $G$ on which every $\mathsf{GNN}_{mp}^n$ of width $w$ requires depth at least $d = \Omega(n/(w \log n) + \delta_G)$ to compute the graph diameter $\delta_G$ and $d = \Omega(\sqrt{n}/(w \log n) + \delta_G)$ to approximate the graph diameter and girth within a factor of $3/2$ and $2$, respectively.*

Term $\delta_G$ appears in the lower bounds because both estimation problems require global information. Further, approximating the diameter within a $3/2$ factor seems to be simpler than computing it. Yet, in both cases, one cannot achieve this using a $\mathsf{GNN}_{mp}^n$ whose capacity is constant. As a final remark, the graphs giving rise to the lower bounds of Corollary 4.5 have constant diameter and $\Theta(n^2)$ edges. However, similar bounds can be derived also for graphs with $O(n \log n)$ edges (Abboud et al., 2016). For the case of exact computation, the lower bound is explained in Appendix C.

## 5 Empirical evidence

This section aims to empirically test the connection between the capacity $dw$ of a $\mathsf{GNN}_{mp}$, the number of nodes $n$ of its input, and its ability to solve a given task. In particular, I considered the problem of 4-cycle classification and tasked the neural network with classifying graphs based on whether they contained a cycle of length four. Following the lower bound construction described in Appendix A, I generated five distributions over graphs with $n \in (8, 16, 24, 32, 44)$ nodes and an average diameter of $(4, 6, 8, 9, 11)$, respectively (this was achieved by setting $p \in (6, 8, 10, 12, 14)$, see Figure 3a). For each such distribution, I generated a training and test set consisting respectively of 1000 and 200 examples. Both sets were exactly balanced, i.e., any example graph from the training and test set had exactly 50% chance of containing a 4-cycle.

The experiment aimed to evaluate how able were $\mathsf{GNN}_{mp}$ of different capacities to attain high accuracy on the test set. To this end, I performed grid search over the hyperparameters $w \in (2, 10, 20)$ and

---

[2]Unless P=NP.

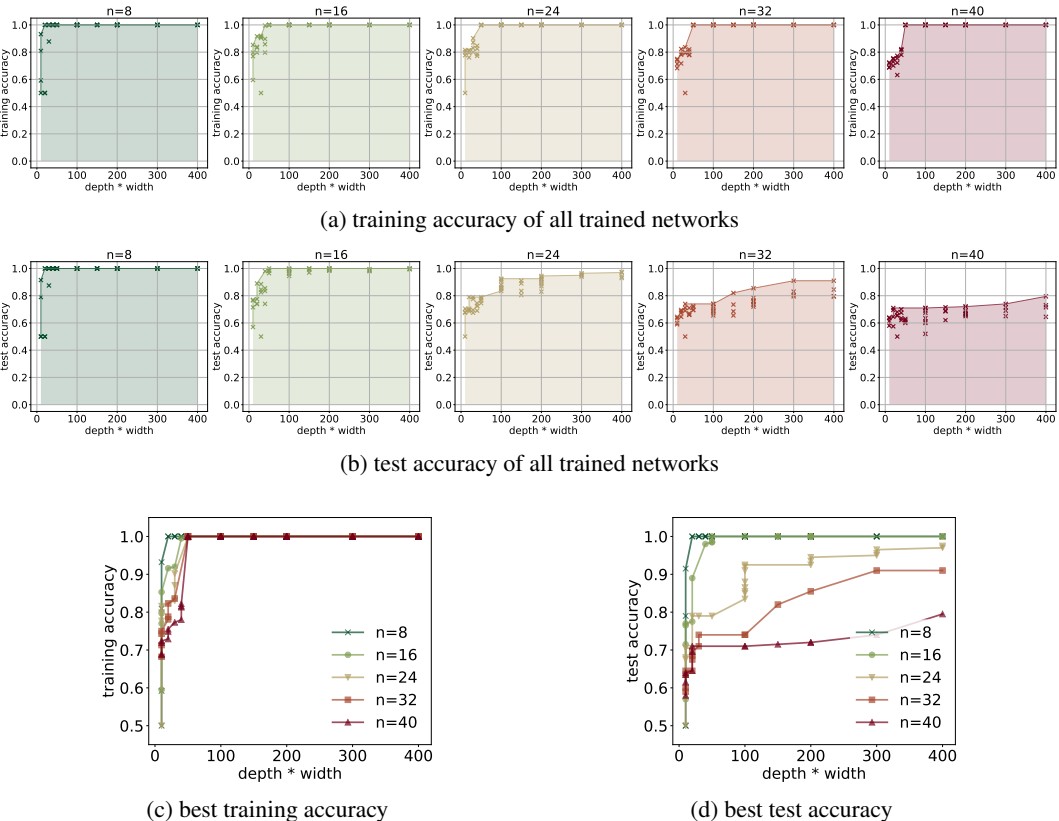

Figure 1: Accuracy as a function of $\mathsf{GNN}_{\mathsf{mp}}$ capacity $dw$ and $n$. (Best seen in color.)

$d \in (5, 10, 20, 15)$. To reduce the dependence on the initial conditions and training length, for each hyperparameter combination, I trained 4 networks independently (using Adam and learning rate decay) for 4000 epochs. The $\mathsf{GNN}_{\mathsf{mp}}$ chosen was that proposed by Xu et al. (2018), with the addition of residual connections—this network outperformed all others that I experimented with.

It is important to stress that empirically verifying lower bounds for neural networks is challenging, because it involves searching over the space of all possible networks in order to find the ones that perform the best. For this reason, an experiment such as the one described above cannot be used to verify[3] the tightness of the bounds: we can never be certain whether the results obtained are the best possible or whether the optimization resulted in a local minimum. In that view, the following results should be interpreted in a qualitative sense. The question that I will ask is: to which extend do the trends uncovered match those predicted by the theory? More specifically, does the ability of a network to detect 4-cycles depend on the relation between $dw$ and $n$?

To answer this question, Figure 1 depicts the training and test accuracy as a function of the capacity $dw$ for all the 240 networks trained (5 distributions $\times$ 3 widths $\times$ 4 depths $\times$ 4 iterations). The accuracy of the best performing networks with the smallest capacity is shown in Figures 1c and 1d. It is important to stress that, based on Weisfeiler-Lehman analyses, anonymous $\mathsf{GNN}_{\mathsf{mp}}$ cannot solve the considered task. However, as it seen in the figures, the impossibility is annulled when using node ids[4]. Indeed, even small neural networks could consistently classify all test examples perfectly (i.e., achieving 100% test accuracy) when $n \leq 16$. Moreover, as the theoretical results predicted, there is a strong correlation between the test accuracy, $dw$ and $n$ (recall that Corollary 4.1 predicts $dw = \tilde{\Omega}(\sqrt{n})$). Figure 1d shows that networks of the same capacity were consistently less accurate on the test set as $n$ increased (even though the cycle length remained 4 in all experiments). It is also

---

[3]This evaluation paradigm, however, can be used to invalidate the theoretical results if one finds that $\mathsf{GNN}_{\mathsf{mp}}$ of small depth and width can solve all lower bound problem instances.

[4]The considered graphs featured the same node set (with different edges) and a one-hot encoding of the node-ids was used as input features.

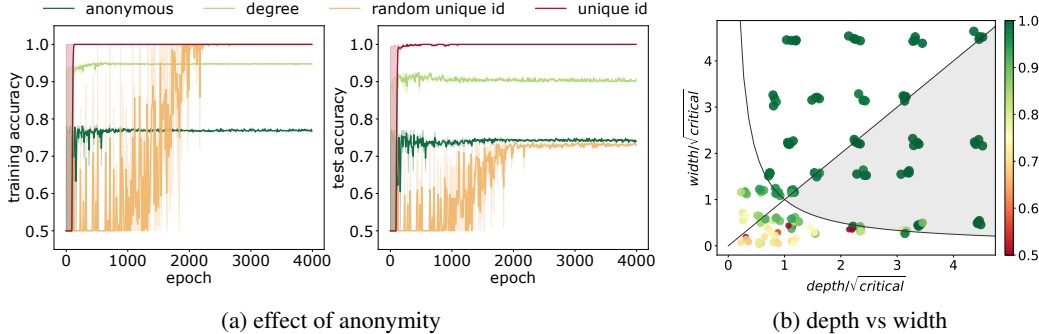

(a) effect of anonymity         (b) depth vs width

Figure 2: (a) GNNs are significantly more powerful when given discriminative node attributes. (b) Test accuracy indicated by color as a function of normalized depth and width. Points in highlighted areas correspond to networks with super-critical capacity, whereas the diagonal line separates networks that more deep than wide. (For improved visibility, points are slightly perturbed. Best seen in color.)

striking to observe that even the most powerful networks considered could not achieve a test accuracy above 95% for $n > 16$; for $n = 40$ their best accuracy was below 80%.

**Effect of anonymity.** Figure 2a plots example training and test curves for $\mathsf{GNN}_{mp}$ trained with four different node attributes: no attributes (anonymous), a one-hot encoding of the node degrees (degree), a one-hot encoding of node ids (unique id), and a one-hot encoding of node ids that changed across graphs (random unique id). It can be clearly observed that there is a direct correlation between accuracy and the type of attributes used. With non- or partially-discriminative attributes, the network could not detect cycles even in the training set. The cycle detection problem was solved exactly with unique ids, but when the latter were inconsistently assigned, the network could not learn to generalize.

**Exchangeability of depth and width.** Figure 2b examines further the relationship between depth, width, and test accuracy. This time, networks were separated depending on their depth and width normalized by the square root of the "*critical capacity*". For each $n$, the critical capacity is the minimum $dw$ of a network that was able to solve the task on a graph of $n$ nodes—here, solving amounts to a test accuracy above 95%. In this way, a network of depth $d$ and width $w$ tested on $n$ nodes corresponds to a point positioned at $x = d/\sqrt{\text{critical}}, y = w/\sqrt{\text{critical}}$ and no network positioned at $xy < 1$ can solve the task (non-highlighted region in the bottom left corner). As seen, there is a crisp phase transition between the regime of under- and super-critical capacity: almost every network meeting the condition $dw \geq \text{critical}$ was able to solve the task, irrespective of whether the depth or width was larger. Note that, the exchangeability of depth and width cannot be guaranteed by the proposed theory which asserts that the condition $dw = \tilde{\Omega}(\sqrt{n})$ is necessary—but not sufficient. The empirical results however do agree with the hypothesis that, for 4-cycle classification, depth and width are indeed exchangeable.

# 6 Conclusion

This work studied the expressive power of graph neural networks falling within the message-passing framework. Two results were derived. First, sufficient conditions were provided such that $\mathsf{GNN}_{mp}$ can compute any function computable by a Turing machine with the same connected graph as input. Second, it was discovered that the product of a $\mathsf{GNN}_{mp}$'s depth and width plays a prominent role in determining whether the network can solve various graph-theoretic problems. Specifically, it was shown that $\mathsf{GNN}_{mp}^n$ with $dw = \tilde{\Omega}(n^\delta)$ and $\delta \in [0.5, 2]$ cannot solve a range of decision, optimization, and estimation problems involving graphs. Overall, the proposed results demonstrate that the power of graph neural networks depends critically on their capacity and illustrate the importance of using discriminative node attributes.

**Acknowledgements.** I thank the Swiss National Science Foundation for supporting this work in the context of the project "*Deep Learning for Graph-Structured Data*" (grant number PZ00P2 179981).

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

## A  GRAPH THEORY DEFINITIONS

The main graph-theoretic terms encountered in this work are:

- *k-cycle detection*: a $k$-cycle is a subraph of $G$ consisting of $k$ nodes, each with degree two. The $k$-cycle detection problem entails determining if $G$ contains a $k$-cycle.

- *Hamiltonian cycle*: a cycle of length $n$

- *(minimum) spanning tree*: a spanning tree is a tree subgraph of $G$ consisting of $n$ nodes. The minimum spanning tree problem entails finding the spanning tree of $G$ of minimum weight (the weight of a tree is equal to the sum of its edge weights).

- *(minimum) cut*: a cut is a subgraph of $G$ that when deleted leaves $G$ disconnected. The minimum cut problem entails finding the cut of minimum weight (the weight of a cut is equal to the sum of its edge weights).

- *s-t cut*: a subgraph of $G$ such that removing all subgraph edges from $G$ will leave the nodes $s$ and $t$ of $G$ disconnected.

- *(shortest) path*: a simple path is subgraph of $G$ where all nodes have degree 2 except from the two endpoint nodes whose degree is one. The shortest path problem entails finding the simple path of minimum weight that connects two given nodes (the weight of a path is equal to the sum of its edge weights).

- *(maximum) independent set*: an independent set is a set of nodes in a graph no two of which are adjacent. The maximum independent set problem entails finding the independent set of maximum cardinality.

- *(minimum) vertex cover*: a vertex cover of $G$ is a set of nodes such that each edge of $G$ is incident to at least one node in the set. The minimum vertex cover problem entails finding the vertex cover of minimum cardinality.

- *(perfect) coloring*: a coloring of $G$ is a labeling of the nodes with distinct colors such that no two adjacent nodes are colored using same color. The perfect coloring problem entails finding a coloring with the smallest number of colors.

- *diameter*: the diameter $\delta_G$ of $G$ equals the length of the longest shortest path.

- *girth*: the girth of $G$ equals the length of the shortest cycle. It is infinity if no cycles are present.

## B  DEFERRED PROOFS

### B.1  PROOF OF THEOREM 3.1

The claim is proven by expressing the state of node $v_i$ in the two models in the same form. It is not difficult to see that for each layer of the $\mathsf{GNN}^n_{mp}$ one has

$$x_i^{(l)} = \mathrm{U_P}_\ell\Big( \sum_{v_j \in \mathcal{N}_i^*} m_{i \leftarrow j}^{(\ell)} \Big) \hspace{3cm} \text{(by definition)}$$

$$= \mathrm{U_P}_\ell\Big( \sum_{v_j \in \mathcal{N}_i^*} \mathrm{M_{SG}}_\ell\left( x_i^{(\ell-1)}, x_j^{(\ell-1)}, v_i, v_j, a_{i \leftarrow j} \right) \Big) \hspace{1cm} \text{(substituted } m_{i \leftarrow j}^{(\ell)})$$

$$= \mathrm{A_{GG}}_\ell\left( \left\{ \left( x_i^{(\ell-1)}, x_j^{(\ell-1)}, v_i, v_j, a_{i \leftarrow j} \right) : v_j \in \mathcal{N}_i^* \right\} \right), \hspace{0.5cm} \text{(from (Xu et al., 2018, Lemma 5))}$$

where $\mathrm{A_{GG}}_\ell$ is an *aggregation function*, i.e., a map from the set of multisets onto some vector space. In the last step, I used a result of Xu et al. Xu et al. (2018) stating that each aggregation function can be decomposed as an element-wise function over each element of the multiset, followed by summation of all elements, and then a final function.

Similarly, one may write:

$$s_i^{(\ell)} = \mathrm{A_{LG}}_\ell^1\left( \left\{ \left( s_{i \leftarrow j}^{(\ell-1)}, a_{i \leftarrow j} \right) : v_j \in \mathcal{N}_i^* \right\}, v_i \right) \hspace{2cm} \text{(by definition)}$$

$$= \mathrm{A_{LG}}_\ell^1\left( \left\{ \left( \mathrm{A_{LG}}_{\ell-1}^2\left( s_j^{(\ell-1)}, v_j \right), a_{i \leftarrow j} \right) : v_j \in \mathcal{N}_i^* \right\}, v_i \right) \hspace{0.8cm} \text{(substituted } s_{i \leftarrow j}^{(\ell-1)})$$

$$= \mathrm{A_{LG}}_\ell\left( \left\{ \left( s_j^{(\ell-1)}, v_i, v_j, a_{i \leftarrow j} \right) : v_j \in \mathcal{N}_i^* \right\} \right),$$

with the last step following by restructuring the input and defining $\mathrm{A_{LG}}_\ell$ as the Turning machine that simulates the action of both $\mathrm{A_{LG}}_\ell^2$ and $\mathrm{A_{LG}}_{\ell-1}^1$.

Since one may encode any vector into a string and vice versa, w.l.o.g. one may assume that the state of each node in **LOCAL** is encoded as a vector $x_i$. Then, to complete the proof, one still needs to demonstrate that the functions

$$\mathrm{A_{GG}}\left( \left\{ (x_i, x_j, v_i, v_j, a_{i \leftarrow j}) : v_j \in \mathcal{N}_i^* \right\} \right) \quad \text{and} \quad \mathrm{A_{LG}}\left( \left\{ (x_j, v_i, v_j, a_{i \leftarrow j}) : v_j \in \mathcal{N}_i^* \right\} \right)$$

are equivalent (in the interest of brevity the layer/round indices have been dropped). If this holds then each layer of $\mathsf{GNN}^n_{mp}$ is equivalent to a round of **LOCAL** and the claim follows.

I first note that, since its input is a multiset, $\mathrm{A_{LG}}_l$ is also an aggregation function. To demonstrate equivalence, one thus needs to show that, despite not having identical inputs, each of the two aggregation functions can be used to replace the other. For the forward direction, it suffices to show that for every aggregation function $\mathrm{A_{GG}}$ there exists $\mathrm{A_{LG}}$ with the same output. Indeed, one may always construct $\mathrm{A_{LG}} = \mathrm{A_{GG}} \circ g$, where $g$ takes as input $\{(x_j, v_i, v_j, a_{i \leftarrow j}) : v_j \in \mathcal{N}_i^*\}$, identifies $x_i$ (by searching for $v_i, v_i$) and appends it to each element of the multiset yielding $\{(x_i, x_j, v_i, v_j, a_{i \leftarrow j}) : v_j \in \mathcal{N}_i^*\}$. The backward direction can also be proven with an elementary construction: given $\mathrm{A_{LG}}$, one sets $\mathrm{A_{GG}} = \mathrm{A_{LG}} \circ h$, where $h$ deletes $x_i$ from each element of the multiset.

## B.2 PROOF OF COROLLARY 3.1

In the LOCAL model the reasoning is elementary (Linial, 1992; Fraigniaud et al., 2013; Seidel, 2015): suppose that the graph is represented by a set of edges and further consider that $\text{ALG}_l$ amounts to a union operation. Then in $d = \delta_G$ rounds, the state of each node will contain the entire graph. The function $\text{ALG}_d^1$ can then be used to make the final computation. This argument also trivially holds for node/edge attributes. The universality of $\text{GNN}_{mp}^n$ then follows by the equivalence of LOCAL and $\text{GNN}_{mp}^n$.

## B.3 PROOF OF THEOREM 4.1

First note that, since the $\text{GNN}_{mp}^n$ and LOCAL models are equivalent, if no further memory/width restrictions are placed, an impossibility for one implies also an impossibility for the other. It can also be seen in Theorem 3.1 that there is a one to one mapping between the internal state of each node at each level between the two models (i.e., variables $x_i^{(l)}$ and $s_i^{(l)}$). As such, impossibility results that rely on restrictions w.r.t. state size (in terms of bits) also transfer between the models.

To proceed, I demonstrate that a depth lower bound in the CONGEST model (i.e., in the LOCAL model with bounded *message size*) also implies the existence of a depth lower bound in the LOCAL model with a bounded *state size*—with this result in place, the proof of the main claim follows directly. As in the statement of the theorem, one starts by assuming that $P$ cannot be solved in less than $d$ rounds when messages are bounded to be at most $b$ bits. Then, for the sake of contradiction, it is supposed that there exists an algorithm $A \in$ LOCAL that can solve $P$ in less than $d$ rounds with a state of at most $c$ bits, but unbounded message size. I argue that the existence of this algorithm also implies the existence of a second algorithm $A'$ whose messages are bounded by $c + \log_2 n$: since each message $s_{j \leftarrow i}^{(l)}$ is the output of a universal Turing machine $\text{ALG}_l^2$ that takes as input the tuple $(s_i^{(l)}, v_i)$, algorithm $A'$ directly sends the input and relies on the universality of $\text{ALG}_{l+1}^1$ to simulate the action of $\text{ALG}_l^2$. The message size bound follows by adding the size $c$ of the state with that of representing the node id ($\log_2 n$ bits suffice for unique node ids). This line of reasoning leads to a contradiction when $c \leq b - \log_2 n$, as it implies that there exists an algorithm (namely $A'$) that can solve $P$ in less than $d$ rounds while using messages of at most $b$ bits. Hence, no algorithm whose state is less than $b - \log_2 n$ bits can solve $P$ in LOCAL, and the width of $\text{GNN}_{mp}^n$ has to be at least $(b - \log_2 n)/p$.

## C AN EXPLANATION OF THE LOWER BOUNDS FOR CYCLE DETECTION AND DIAMETER ESTIMATION

A common technique for obtaining lower bounds in the CONGEST model is by reduction to the *set-disjointness* problem in two-player communication complexity: Suppose that Alice and Bob are each given some secret string ($s_a$ and $s_b$) of $q$ bits. The two players use the string to construct a set by selecting the elements from the base set $\{1, 2, \ldots, q\}$ for which the corresponding bit is one. It is known that Alice and Bob cannot determine whether their sets are disjoint or not without exchanging at least $\Omega(q)$ bits (Kalyanasundaram & Schintger, 1992; Chor & Goldreich, 1988).

The reduction involves constructing a graph that is partially known by each player. Usually, Alice and Bob start knowing half of the graph (red and green induced subgraphs in Figure 3). The players then use their secret string to control some aspect of their private topology (subgraphs annotated in dark gray). Let the resulting graph be $G(s_a, s_b)$ and denote by cut the number of edges connecting the subgraphs controlled by Alice and Bob. To derive a lower bound for some problem $P$, one needs to prove that a solution for $P$ in $G(s_a, s_b)$ would also reveal whether the two sets are disjoint or not. Since each player can exchange at most $O(b \cdot \text{cut})$ bits per round, at least $\Omega(q/(b \cdot \text{cut}))$ rounds are needed in total in CONGEST. By Theorem 4.1, one then obtains a $d = \Omega(q/(w \log n \cdot \text{cut}))$ depth lower bound for $\text{GNN}_{mp}^n$.

The two examples in Figure 3 illustrate the graphs $G(s_a, s_b)$ giving rise to the lower bounds for even $k$-cycle detection and diameter estimation. To reduce occlusion, only a subset of the edges are shown.

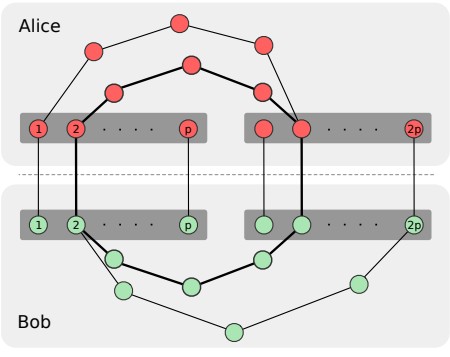 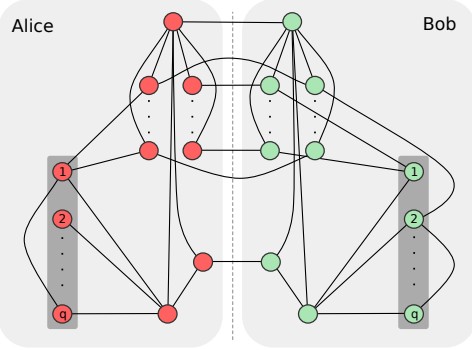

(a) 10-cycle lower bound graph         (b) diameter lower bound graph

Figure 3: *Examples of graphs giving rise to lower bounds.*

(a) In the construction of Korhonen & Rybicki (2018), each player starts from a complete bipartite graph of $p = \sqrt{q}$ nodes (nodes annotated in dark grey) with nodes numbered from 1 to $2p$. The nodes with the same id are connected yielding a cut of size $2p$. Each player then uses its secret (there are as many bits as bipartite edges) to decide which of the bipartite edges will be deleted (corresponding to zero bits). Remaining edges are substituted by a path of length $k/2 - 1$. This happens in a way that ensures that $G(s_a, s_b)$ contains a cycle of length $k$ (half known by Alice and half by Bob) if and only if the two sets are disjoint: the cycle will pass through nodes $t$ and $p + t$ of each player to signify that the $t$-th bits of $s_a$ and $s_b$ are both one. It can then be shown that $n = \Theta(p^2)$ from which it follows that: CONGEST requires at least $d = \Omega(q/(b \cdot \text{cut})) = \Omega(n/(b \cdot p)) = \Omega(\sqrt{n}/b)$ bits to decide if there is a cycle of length $k$; and $\mathsf{GNN}_{mp}^n$ has to have $d = \Omega(\sqrt{n}/(w \log n))$ depth to do the same.

(b) In the construction of Abboud et al. (2016), each string consists of $q = \Omega(n)$ bits. The strings are used to encode the connectivity of subgraphs annotated in dark gray: an edge exists between the red nodes $i$ and $q$ if and only if the $i$-th bit of $s_a$ is one (and similarly for green). Due to the graph construction, the cut between Alice and Bob has $O(\log q)$ edges. Moreover, About et al. proved that $G(s_a, s_b)$ has diameter at least five if and only if the sets defined by $s_a$ and $s_b$ are disjoint. This implies that $d = \Omega(n/(w \log^2 n))$ depth is necessary to compute the graph diameter in $\mathsf{GNN}_{mp}^n$.

# D  THE COST OF ANONYMITY

There is a striking difference between the power of anonymous networks and those in which nodes have the ability to uniquely identify each other, e.g., based on ids or discriminative attributes (see the survey by Suomela (2013)).

To illustrate this phenomenon, I consider a thought experiment where a node is tasked with reconstructing the graph topology in the LOCAL model. In the left, Figure 4 depicts the red node's knowledge after two rounds (equivalent to a $\mathsf{GNN}_{mp}^n$ having $d = 2$) when each node has a unique identifier (color). At the end of the first round, each node is aware of its neighbors and after two rounds the entire graph has been successfully reconstructed in red's memory.

In the right subfigure, nodes do not possess ids (as in the analysis of (Xu et al., 2018; Morris et al., 2019)) and thus cannot distinguish which of their neighbors are themselves adjacent. As such, the red node cannot tell whether the graph contains cycles: after two rounds there are at least two plausible topologies that could explain its observations.

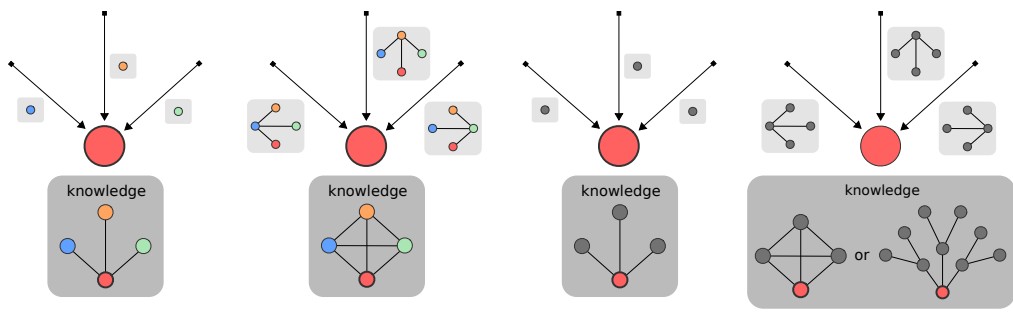

(a) nodes *can* identify each other        (b) nodes *cannot* identify each other (WL test).

Figure 4: *Toy example of message exchange from the perspective of the red node. The arrows show where each received message comes from and the message content is shown in light gray boxes. Red's knowledge of the graph topology is depicted at the bottom.*

