# OpenReview forum: "What graph neural networks cannot learn: depth vs width"
_ICLR.cc/2020/Conference — Accept (Poster)_

### Official Review · AnonReviewer3 · 2019-10-23
**Official Blind Review #3**

**Rating:** 8

**Review:**

Summary

This paper studied the expressive power of graph NNs, specifically, their universality and limitations under the non-anonymous setting, via the theory of distributed computations. For the universality, it proved the Turing completeness of graph NNs if messaging and aggregation functions are sufficiently strong. For the limitation, it characterized the lower bound of width for solving graph-theoretic tasks (such as subgraph detection, subgraph verification, approximate, and exact optimization problems) using graph NNs. The key idea is to reduce the computation model of graph NNs to LOCAL (for Turing completeness) or CONGEST (for limitations), which are well-studied in the literature of distributed computations and use the known results for these models.


Decision
This paper gave us a new approach to analyzing the expressive power of graph NNs. Not only does this paper give new theoretical results, but also it opens the door to a new research direction by bridging the theories of graph NNs and distributed computations. However, I cannot confirm the correctness of the proof of Theorem 3.1 (see Suggestions section). For now, I am tending to accept the paper. But I want to determine the final decision after I am certain that the proof of the theorem is correct.
We can roughly divide existing approaches for studying the expressive power of graph NNs into two. One is to compare the power of discriminating non-isomorphic graph pairs with isomorphism tests such as the WL isomorphism test (Xu et al., 2019). The other one is to theoretically justify the oversmoothing phenomena (Li et al., 2018). The proof techniques the authors used are different from both of the two. It related a graph NN to the computational models LOCAL and CONGEST, and enabled to incorporate the theory of distributed computations. By doing so, the authors successfully derived many lower bounds in a systematic way, proving the effectiveness of their strategy. I think we can expect that a more refined analysis inspired by this approach will appear in the future.

Regarding the Experience Assessment: I have published several papers in graph NNs (4). But I do not know much about the area of the theory of distributed algorithms (1--3).


Suggestions

- Section 3.2
	- Theorem 3.1 proves the equivalence of GNN_n and LOCAL. However, the definition of equivalence is missing. Please write it in the main part, since this theorem is the key result of this paper.
	- I could not find any reference for the Turing completeness of the LOCAL model. Could you add the reference for it?
	- The description of the CONGEST model is only available in the appendix informally (Appendix B.3). Could you write it in the main part?
	- The authors emphasized the importance of the universality and limitation results in the introduction and paragraph after Corollary 3.1. In my opinion, the importance of such tasks is in machine learning community (Cybenko's paper on the universality of MLPs (Cybenko, 1989) is one of the most cited papers in the community). Rather, I think many graph NN researchers who are expected to read this paper are not familiar with the theory of distributed computations. Therefore, I would recommend to use page resources to explain the basic concepts of the distributed computation theory.


Questions

- Is there any existing work which tries to solve the graph theoretical tasks using graph NNs? If there is, can the theorems in this paper give explanations for the results?


[Cybenko, 1989] Cybenko, George. Approximation by superpositions of a sigmoidal function. Mathematics of control, signals and systems 2.4 (1989): 303-314.
[Li et al., 2018] Qimai Li, Zhichao Han, and Xiao-Ming Wu. Deeper Insights into Graph Convolutional Networks for Semi-Supervised Learning. In Proceedings of the 32nd AAAI Conference on Artificial Intelli- gence, pp. 3538–3545, 2018.
[Xu et al., 2019] Keyulu Xu, Weihua Hu, Jure Leskovec, and Stefanie Jegelka. How powerful are graph neural networks? In International Conference on Learning Representations, 2019.


**Experience Assessment:**

I have read many papers in this area.

**Review Assessment: Checking Correctness Of Derivations And Theory:**

I assessed the sensibility of the derivations and theory.

**Review Assessment: Checking Correctness Of Experiments:**

N/A

**Review Assessment: Thoroughness In Paper Reading:**

I read the paper at least twice and used my best judgement in assessing the paper.

---

> ### Author Response · Authors · 2019-11-14
> **Reply to Review #3**
>
> Let me start by thanking you for your comments and suggestions. I share your opinion that this new connection with distributed computing opens up new possibilities for understanding the expressive power of GNN---and, as you predicted, I am very excited to continue developing this direction.
>
>
> S1. Add a definition of equivalence.
>
> Thank you for this suggestion. In the updated version, I have updated the statement of Theorem 3.1 to explain that the two models can be parametrized such that the binary representation of their node states is identical at every layer/round. I believe that making the notion of equivalence explicit helps significantly to clarify the proof of Theorem 3.1.
>
> S2. Provide a reference for the Turing completeness of the LOCAL model.
>
> Due to its direct proof (LOCAL is a network of processors running Turing machines, thus given unbounded memory and message length each processor sees the entire input after diameter rounds), the Turing completeness is considered a folk theorem---it is known and considered to have established status, but has never been published in complete form.
>
> The first mention I could find was in the seminal paper [Locality in distributed graph algorithms. Linial. SIAM Journal on Computing, 1992.] : “It is clear that the present model (meaning LOCAL) allows us to compute every function of G in time (equivalent to rounds) O(diameter(G)). After this amount of time every processor obtains a complete knowledge of both G and ID.”
>
> To quote two more recent works:
> * [Anonymous Distributed Computing: Computability, Randomization, and Checkability. Ph.D. thesis of Jochen Seidel, 2015, page 1]: “Computability in (LOCAL) networks (..) is Turing machine-equivalent if the nodes are equipped with unique IDs.”
> * [Towards a Complexity Theory for Local Distributed Computing, by Pierre Fraigniaud, Amos Korman, and David Peleg in Journal of the ACM (JACM), 2013, page 2]: “Note that in the LOCAL model, every decidable problem can be solved in a number of rounds equal to the diameter of the input graph”
>
> As per your suggestion, I have added the above references to the manuscript.
>
> S3. Description of the CONGEST model
>
> I added a short explanation at the end of 3.1 explaining that the CONGEST model is identical to the LOCAL model (explained in Section 3.1), with the only difference that the message size is bounded to be at most $b$ bits. There are no other differences between the models so I am not sure what else should be explained.
>
> S4. I would recommend using page resources to explain the basic concepts of the distributed computation theory.
>
> I have made certain changes to that effect, but with the additional changes suggested by the reviewers, the space remaining to give background information on distributed computing theory is still quite limited.
>
> Q1. Is there any existing work that tries to solve the graph-theoretical tasks using graph NNs? If there is, can the theorems in this paper give explanations for the results?
>
> There are two relevant papers that I am aware of:
> * [Mincut pooling in Graph Neural Networks. FM Bianchi et al. 2019] partly attempts to solve a min-cut optimization problem in order to find pooling sets. Unfortunately, its experiments did not focus on the pure min-cut problem but used it as a proxy for pooling.
> * [Relational inductive biases, deep learning, and graph networks. P Battaglia et al. 2018.] considers the shortest s-t path problem. The considered network, however, does not belong to the message passing framework.
>
> Nevertheless, I was able to empirically evaluate the bounds myself. Specifically, I have added in Appendix D of the updated manuscript empirical results for the fundamental problem of cycle detection. For a summary of the main insights of these results, please refer to the answer to reviewer #2.

---

### Official Review · AnonReviewer2 · 2019-10-23
**Official Blind Review #2**

**Rating:** 6

**Review:**

This paper studies theoretical properties of GNN in particular their expressive power. There are many recent works on this topic and the 2019 ICLR paper 'How Powerful are Graph Neural Networks?' is the closes related to this paper. In the 2019 paper connects GNN with the  Weisfeiler-Lehman graph isomorphism test in theoretical computer science. This paper makes a connection between GNN and the locality notion developed in distributed computing.
This connection is rather obvious and GNN being particular local algorithms, their expressive power is at least as limited as the expressive power of local algorithms. In this paper, results in distributed computing are reformulated in a GNN framework mapping the number of rounds required by a local algorithm to the depth of the GNN in order to solve a given graph problem in a worst case scenario.
In my opinion, this paper is rather incremental. In order to improve it, it would be nice to see experiments supporting the theoretical results in section 4. Here it is not clear at all if the bounds given are tight in practice.

###
The authors added experiments supporting their theoretical results. I am upgrading my rating to weak accept.

**Experience Assessment:**

I have read many papers in this area.

**Review Assessment: Checking Correctness Of Derivations And Theory:**

I assessed the sensibility of the derivations and theory.

**Review Assessment: Checking Correctness Of Experiments:**

N/A

**Review Assessment: Thoroughness In Paper Reading:**

I read the paper at least twice and used my best judgement in assessing the paper.

---

> ### Author Response · Authors · 2019-11-14
> **Reply to Review #2**
>
> Thank you for your review. I answer your two comments below:
>
> C1. This paper is rather incremental.
>
> I respectfully disagree with the assessment. Despite technical simplicity (which I believe is a good thing), this work brings 3 main novel insights to the GNN community:
>
> a. This work is the first to provide sufficient universality conditions for message-passing GNN and goes directly in contrast to Weisfeiler-Lehman-based results (as explained in the paper, WL lower bounds only hold for the anonymous case). Universality was shown previously for special variants of GNN (see references in the paper), but not for the message passing framework.
>
> b. This work shows that the solution of a large number of tasks hinges on the product depth*width being larger than a function of $n$. Such results were previously unknown for GNN and are derived based on a novel technique connecting communication complexity in CONGEST with GNN_{MP} width. I am also not aware of depth*width type results in the context of deep learning, but I might be mistaken.
>
> c. A direct implication is that the number of parameters of a GNN_{MP} (which are $\Omega(depth*width)$ when UP is a linear layer followed by a non-linearity and  $\Omega(depth*width^2)$ when UP is a feed-forward neural network with one hidden layer) cannot, in general, be independent of $n$. This refutes the common intuition that, with graph convolution, simple tasks (e.g., 4-cycle detection or connectivity verification) can be solved with $O(1)$ parameters.
>
> Based on the novelty and importance of these insights, I argue that this work carries a strong contribution.
>
> C2. It would be nice to see experiments supporting the theoretical results in section 4. Here it is not clear at all if the bounds given are tight in practice.
>
> Thank you for your constructive comment. Let me first argue that non-trivial lower bounds can be relevant even if not tight. That is because they give necessary conditions for the completion of a task: no matter which training procedure or architecture variant is used, it is impossible to get the correct solution for every input when the condition is violated. Further, as mentioned above, the lower bounds presented here are non-trivial as they defy common intuition (e.g., for the case of cycle detection).
>
> Nevertheless, I fundamentally agree that empirical evidence would make the paper stronger. To that end, I have added in Appendix D experimental results demonstrating the connection between $dw$, $n$, and the ability of a GNN_{MP} to detect whether the graph contains a 4 cycle.
>
> These results corroborate the following points of the theory:
> * By showing that a 4-cycle detection can be perfectly solved (100% test accuracy) by a message-passing GNN, they confirm the paper’s claims that Weisfeiler-Lehman-type analyses are overly pessimistic for non-anonymous networks (according to WL-based results this task should be impossible).
>
> * As the theoretical results predicted, a strong correlation is found between the test accuracy, $d w$ and $n$. Figure 2d shows that networks of the same capacity $dw$  were consistently less accurate on the test set as $n$ was increased (even though the cycle length remained 4 in all experiments). It was also striking to observe that even the most powerful networks considered could not achieve a test accuracy above 95% for $n>16$. For $n=40$, the best test accuracy was below 80%.
>
> * In addition, it is determined that there exists a crisp phase transition between the regime of under- and super-critical capacity (the smallest value $dw$ for which a network can solve the task for a particular $n$). Almost every network satisfying the condition $d * w \geq critical$ was able to solve the task, independently of whether the depth or width was larger.
>
> The description of the experiments and results can be found in Appendix D of the revised manuscript. I urge the reviewer to consider them in detail.
>
> Beyond 4-cycle classification, I have also produced similar evidence for two additional connectivity verification tasks (not included in the revised manuscript). As you might imagine, to be thorough, the experiments are quite involved, both in terms of their description and analysis. For these reasons, my original intention was to write an accompanying manuscript that provides an in-depth technical exposition (rather than cramming them in the appendix of the submitted paper). In view of your comments, I decided to share a subset of these results --- so that I can convince you that the trends predicted by the lower bounds are in fact predictive of reality. Nevertheless, I am not certain whether adding them to the (already lengthy) appendix is the best way to proceed. I would appreciate it if the three reviewers and the area chair could comment on their preferred course of action: should I keep the k-cycle results in the appendix or should I instead incorporate them to a dedicated manuscript that performs an in-depth evaluation?

---

### Official Review · AnonReviewer1 · 2019-10-25
**Official Blind Review #1**

**Rating:** 8

**Review:**

This paper studies impossibility results of GNN in the worst-case sense. In particular, it reduces GNN to a distributed computing model CONGEST and adapt the impossibility result from distributed computing to GNNs. The impossibility results show that for certain problems, e.g., subgraph detection, there exists a graph such that GNN can not solve the problem unless the product of a GNN’s depth and width exceeds (a function of) the graph size.

I am not an expert of distributed computing and I did not check all the proofs thoroughly. But I do think this paper provides a solid contribution to broaden the community’s understanding about what the limitations of GNNs are. Overall, I tend to accept it and would like to increase the score based on authors’ feedback.

Pros:

1, I like the contribution of the paper which tries to build connections between GNNs and distributed computing models. From the perspective of computation, GNNs and distributed algorithms do share a lot of similarities. Therefore, some algorithm design choices in distributed computing would shed some light on designing novel GNNs. This may open a new direction for the community.

2, The depth and width dependency results are novel in the context of GNNs.

Cons & Questions & Suggestions:

1, Since these impossibility results for a certain subclass of GNNs are in the worst-case sense, it is not clear how it would be useful for practical machine learning problems. Some discussion along this line would be very helpful.

2, It would be great to discuss the relationship between the Turing universality and the universality of function approximation studied in [1].

3, For people who have no background of distributed computing, it would be great to describe CONGEST before going to the impossibility results reduced from CONGEST to GNNs.

4, I do not recommend authors to refer to the computation model 1 as GNN. You could name it as MPNN in order to make the claim more accurate. GNN in general has a few variants which does not fall into this category and could have higher capacity than MPNN. For example, the authors claim that “graph neural networks always sum received messages before any local computation”. However, this is not true in GraphSAGE [2] where the aggregation is a LSTM rather than a simple sum. It makes the model resemble more to the computational model 2. Recent spectral graph convolutional networks [3,4] leverages Krylov subspace methods to compute approximated eigenvalues and eigenvectors of the graph Laplacian which are further used to compute long-range propagation / high-power Laplacian to improve representation power. The results on depth may not hold for these models any more since one layer graph convolution could aggregate multi-hop information. Therefore, being more specific on the model class would make the conclusion more accurate. It would be great to discuss these models separately from the computation model 1.

[1] Chen, Z., Villar, S., Chen, L. and Bruna, J., 2019. On the equivalence between graph isomorphism testing and function approximation with GNNs. arXiv preprint arXiv:1905.12560.
[2] Hamilton, W., Ying, Z. and Leskovec, J., 2017. Inductive representation learning on large graphs. In Advances in Neural Information Processing Systems (pp. 1024-1034).
[3] Liao, R., Zhao, Z., Urtasun, R. and Zemel, R.S., 2019. Lanczosnet: Multi-scale deep graph convolutional networks. arXiv preprint arXiv:1901.01484.
[4] Luan, S., Zhao, M., Chang, X.W. and Precup, D., 2019. Break the Ceiling: Stronger Multi-scale Deep Graph Convolutional Networks. arXiv preprint arXiv:1906.02174.

======================================================================================================

The response from authors address most of my concerns. I improved the score.


**Experience Assessment:**

I have published in this field for several years.

**Review Assessment: Checking Correctness Of Derivations And Theory:**

I assessed the sensibility of the derivations and theory.

**Review Assessment: Checking Correctness Of Experiments:**

N/A

**Review Assessment: Thoroughness In Paper Reading:**

I read the paper at least twice and used my best judgement in assessing the paper.

---

> ### Author Response · Authors · 2019-11-14
> **Reply to Review #1**
>
> Thank you for your constructive evaluation and comments. Below, I address your comments one by one.
>
> C1. Since these impossibility results for a certain subclass of GNNs are in the worst-case sense, it is not clear how it would be useful for practical machine learning problems. Some discussion along this line would be very helpful.
>
> Indeed, the lower bounds discussed are in a worst-case sense and thus might not hold for datasets that do not contain bad instances (i.e., problematic graphs giving rise to the lower bound). The benefit of worst-case analyses is that they give us an understanding of the limits of the capacity of a (learning) machine. For instance, here one realizes that certain problems that can appear trivial from a classical point of view (e.g., connectivity verification, diameter estimation, short cycle detection, shortest-path computation) are, in certain instances, difficult for a message-passing GNN (meaning that the number of parameters have to be strongly dependent on the number of nodes of the graph, even when the graph diameter isn’t). At the same time, I agree with the reviewer that different types of lower bounds could also be useful and I would be interested to work further in this direction. I have added a sentence in the updated introduction to point out that the discovery of non-worst-case depth-vs-width lower bounds remains an open problem.
>
> C2. It would be great to discuss the relationship between the Turing universality and the universality of function approximation studied in [1].
>
> This is an excellent recent reference. The work shows that there is an equivalence between graph isomorphism testing and the approximation of permutation-invariant functions. Since Turing universality is more powerful that universal approximation, the above also implies that “Turing universal classes of functions are also GIso-discriminating”. The reverse statement is not a direct consequence, but I suspect that it could be proven (based on a modification of Lemma 2 in [1]). I added a short remark about this in the updated introduction.
>
> C3. For people who have no background in distributed computing, it would be great to describe CONGEST before going to the impossibility results reduced from CONGEST to GNNs.
>
> I added a short explanation at the end of 3.1 explaining that the CONGEST model is identical to the LOCAL model (explained in Section 3.1), with the only difference that the message size is bounded to be at most $b$ bits.  There are no other differences between the models, so I am not sure what else should be explained.
>
> C4. (paraphrased) Not every GNN falls into the message passing framework (e.g., see [2,3,4]). Rename model 1 to clarify that it refers to a specific variant of graph neural networks.
>
> Following your recommendation, I have renamed GNN (as GNN$_{\text{MP}}$) to emphasize that the results only apply for those networks that fall within the message-passing framework. I have also added some examples of GNNs for which the theory doesn’t apply.
>
> Let me also briefly point out some relevant connections that I have identified. As you pointed out, in their natural form [2-4] do not fall within model 1. In some instances, however, the difference is not significant, due to the following observation: certain networks might not fall within the message-passing framework, but there still exists a message-passing GNN which gives the same output.
> * In GraphSage, the LSTM aggregator can be substituted by a sum-aggregator without loss of generality (this follows from (Xu et al 2018)).
> * In multi-hop spectral methods, a subcase of which seem to be [3,4], each GNN layer implements a graph filter and thus it cannot be written in the form of model 1. However, whenever the filters can be expressed as polynomials of some graph operator, such as the Laplacian or the adjacency matrix, a single polynomial spectral layer of order k can be decomposed as a sequence of k linear message passing layers (by the Stone-Weierstrass theorem, any graph filter with continuous spectral response can be approximated arbitrarily well by a polynomial graph filter). Thus, under this line of reasoning, spectral methods can be thought of as very deep message passing graph neural networks, but with simple layers.
>
> In light of the space limitations, I decided to not include the above discussion in the paper. I can add a remark in the appendix if you feel it is essential.

---

### Public Comment · ~Ryoma_Sato1 · 2019-11-08
**A related work on the connection between GNNs and local algorithms**

Hi, the analysis of the depth and width of GNNs is interesting, and I really enjoyed reading it. It is good for this paper to refer to the following paper because it first pointed out the connection of GNNs to distributed local algorithms. Especially, it gave a theoretical consideration about the limit of the ability of GNNs in terms of approximation ratios by pointing out the connections between GNNs and local algorithms.

[1] Ryoma Sato, Makoto Yamada, Hisashi Kashima. Approximation Ratios of Graph Neural Networks for Combinatorial Problems. NeurIPS 2019. https://arxiv.org/abs/1905.10261

Thank you for your attention.

---

> ### Author Response · Authors · 2019-11-14
> **Relation to recent related work**
>
> Thank you for pointing out this interesting recent reference. As will be explained in the following, though the two papers share some common ideas, ultimately their contributions are different.
>
> For easy reference, I define:
> [1] Ryoma Sato, Makoto Yamada, Hisashi Kashima. Approximation Ratios of Graph Neural Networks for Combinatorial Problems. NeurIPS 2019. https://arxiv.org/abs/1905.10261
> [2] Anonymous author. What graph neural networks cannot learn: depth vs width. Submitted to ICLR 2020
>
> An important note: Before I discuss the main similarities and differences, I would like to point out that based on the ICLR regulations the two papers should be considered as parallel work: [1] was published at Neurips 2019 less than 1 month prior to the ICLR deadline. Moreover, the two works appeared on arxiv within ~2 months of each other (24 May 2019 vs 6 July 2019). In both cases, the dates differ less than four months --- meeting ICLR’s requirement for parallel work (https://twitter.com/Aistats2020/status/1193586342606884867?s=20).
>
> Common ideas:
> 1. Both papers identify a connection between GNN and LOCAL.
> 2. Both papers consider lower bounds for combinatorial problems. [1] focuses on the minimum dominating set, minimum vertex cover, and maximum matching. [2] considers a large number of problems relating to verification, detection, optimization, and estimation (the problems are too numerous to list here).
>
> Main differences:
> 1. The GNNs considered in the two works are different: [1] focuses on custom port-numbered and color-enhanced networks. These are strictly less powerful than the GNN message-passing model GNN_{MP} considered in [2]. Indeed, the GNNs presented in [1] are not universal, whereas GNN_{MP} is.
>
> 2. The lower bounds presented in [2] are more refined:
> 2a. First, the lower bounds presented in the two works emerge from different limitations of the studied networks. To see the difference, notice that all bounds presented in [1] are invalid in the setting of [2] (this follows directly by the universality of GNN_{MP}).
>
> 2b. Second, rather than saying that something is impossible under any instantiation of the network, in [2] explicit conditions are given on the necessary relation between the depth and width of a neural network together with the size of the input graph. This yields more insightful bounds that connect the capacity of a network to solve a problem with its hyperparameters and the properties of its input. Also, rather than focusing on individual cases, [2] presents a generic methodology that allows one to systematically repurpose lower bounds from distributed computing (Theorem 4.1).
>
> I have added a short discussion about [1] in the updated version of the manuscript.

---

### Decision · Program_Chairs · 2019-12-19

**Decision:**

Accept (Poster)

**Comment:**

This paper provides a theoretical background for the expressive power of graph convolutional networks. The results are obviously useful, and the discussion went in the positive way. All reviewers recommend accepting, and I am with them.